# HP-CagA+ Regulates the Expression of CDK4/CyclinD1 via *reg3* to Change Cell Cycle and Promote Cell Proliferation

**DOI:** 10.3390/ijms21010224

**Published:** 2019-12-28

**Authors:** Bin Liu, Xiaokang Li, Fuze Sun, Xiaoling Tong, Yanmin Bai, Kairang Jin, Lin Liu, Fangyin Dai, Niannian Li

**Affiliations:** 1Teda Institute of Biological Sciences & Biotechnology, Nankai University, Tianjin 300457, China; Liubin123jinjin@sina.com; 2The Key Laboratory of Molecular Microbiology & Technology, Ministry of Education, Nankai University, Tianjin 300071, China; 3Tianjin Key Laboratory of Microbial Functional Genomics, Nankai University, Tianjin 300457, China; 4School of Electronics and Information, Shanghai Electric Power University, Shanghai 200090, China; 5State Key Laboratory of Silkworm Genome Biology, Key Laboratory of Sericultural Biology and Genetic Breeding, Ministry of Agriculture, College of Biotechnology, Southwest University, Chongqing 400715, China; 6School of Life Sciences, Nankai University, Tianjin 300071, China

**Keywords:** gastric cancer, *reg3*, HP-CagA+, cell cycle, CDK4/CyclinD1

## Abstract

Previous studies have shown that regeneration gene 3 (*reg3*) is significantly expressed in gastric mucosa tissues with *Helicobacter pylori* (HP) cytotoxin-associated gene A (CagA)-positive (HP-CagA+). CagA-positive HP increases the risk of gastric cancer. The purpose of this study was to investigate the correlation between *reg3* and HP-CagA+ and explore the effects of *reg3* on the proliferation of gastric cancer cells and the development of tissues and organs. We analyzed the expression of reg3 in human tissues and organs. The results showed that *reg3* expression in gastric tissues was significantly higher than that in other tissues and organs. In addition, *reg3* influenced the prognosis of gastric, lung, and ovarian cancers. Immunohistochemical analysis indicated that the expression of *reg3* and CagA in cancerous tissues was higher than that in adjacent tissues. HP-CagA^+^ infection of gastric cancer cells promotes reg3 expression, suggesting that *reg3* may be a target gene of CagA in gastric cancer, which together affects the formation and development of gastric cancer. *reg3* and CagA promote cell proliferation, and then affect the development of mouse tissues and organs by regulating G1/S phase transition of the cell cycle via the formation of the cell cycle-dependent complex CDK4/CyclinD1. This is the first study that shows the influence of CagA on the cell cycle and induction of cell proliferation by promoting *reg3* expression.

## 1. Introduction

Gastric cancer is one of the most common malignant tumors in the world. The five-year survival rate is less than 40%, thereby making it a serious threat to human health [1]. At present, there is still no effective treatment for gastric cancer [2]. Therefore, it is necessary to further explore the mechanism of the occurrence and development of gastric cancer, and to prevent the incidence of gastric cancer. The potential role of Reg in tumors, particularly in digestive system tumors, has been recently investigated [3,4]. It is currently determined that the Reg family proteins belong to the C-type lectin family, depending on the sequence characteristics of the protein [3]. These were estimated to have about 120 amino acids constituting a calcium-dependent carbohydrate recognition domain (CRD) [5], which are present in serum [6], extracellular matrix [7], and cell membrane [8], and C-type lectin on the cell membrane can participate in swallowing, cell adhesion, or humoral immune response [9,10].

Studies have shown that *Helicobacter pylori* infection is an important cause of gastric cancer [11,12]. Epidemiology shows that the major protein toxic factors carried by *H. pylori* strains is CagA, which can increase the risk of gastric cancer [13,14]. In human gastric mucosa tissues, the expression of Reg3 was significantly increased in the presence of CagA compared to CagA-negative *H. pylori*-infected individuals [15]. RT-PCR (reverse transcription-polymerase chain reaction) demonstrated a significant increase in the expression of rat *reg3* in CagA-positive HP-infected gastric mucosal biopsies but not in CagA-negative tissues [16]. *reg3* may act as a targeting molecule for CagA in gastric epithelial cells [15,16]. CagA-mediated SHP signaling can lead to abnormalities in epithelial cell polarity [17,18]. In particular, the transformation of potentially carcinogenic epithelial cells into mesenchymal cells (EMT) is attributed thereto [19]. Studies involving transgenic mice have further supported the hypothesis that CagA may be a potential carcinogenic factor [20]. However, how reg3 interacts with CagA in the stomach and affects the occurrence and development of gastric cancer, as well as how to regulate downstream genes to induce tumors, have not been studied to date. Thus, the purpose of this study was to explore the carcinogenic mechanism of Reg3 and CagA and to provide some basis for cancer treatment.

## 2. Results

### 2.1. reg3 Is Highly Expressed in Gastric Cancer and Affects the Prognosis of Patients

*reg3* is a member of the regenerative gene family and has attracted much attention in relation to inflammatory wounds and digestive tract tumors. The expression of *reg3* in three common tumors, including gastric, lung, and ovarian cancer, and the effect of the *reg3* on survival and prognosis of patients were analyzed using TCGA data (Figure 1A,B). The analysis showed that the expression level of *reg3* in these three tumor tissues was higher than that in normal tissues. In addition, the upregulation of this gene is associated with poor prognosis. Meanwhile, the human protein database was first used to analyze the expression of *reg3* in human tissues and organs (www.proteinatlas.org) (Figure 1C). The results indicated that the level of *reg3* expression in gastric tissues, including the stomach, duodenum, and small intestine, was relatively higher than other the tissues and organs. *reg3* is closely related to the occurrence of gastric cancer, and is an important growth gene and regulatory target in gastric cancer pathogenesis. The results of our analysis prove that *reg3* plays an important role in the development of the stomach and influences the formation and development of gastric cancer.

### 2.2. CagA Promotes Cell Proliferation by Regulating reg3

*Helicobacter pylori* (HP) infection is an important cause of gastric cancer. The CagA carried by the HP can increase the risk of gastric cancer. Immunohistochemistry was used to detect the expression of CagA and reg3 in gastric cancer (Figure 2A,B). The results show that the expression level of CagA and *reg3* in cancerous tissues is higher than that in the adjacent tissues. Reg3 was upregulated in gastric cancer cell lines, and the effect of Reg3 on cell proliferation was detected by the MTT assay and cell counting (Figure 2C). The results show that the upregulation of *reg3* can promote cell proliferation. In addition, CagA-positive *H. pylori* (HP-CagA^+^) was used to infect gastric cancer cells to detect changes in cell proliferation (Figure 2D). The results show that as infection time is prolonged, cell proliferation is accelerated. RT-PCR analysis indicated that the expression of *reg3* in gastric mucosa infected by CagA-positive *H. pylori* significantly increased in mice while the expression of Reg3 in gastric mucosa infected by CagA-negative *H. pylori* was comparable to that in normal gastric mucosa. In this study, CagA and Reg3 were upregulated in gastric cancer tissues and promote gastric cancer cell proliferation. However, the relationship between CagA and Reg3 expression is unclear. After HP-CagA^+^ was transfected into gastric cancer cells, reg3 expression was detected at 0, 24, and 48 h (Figure 2E,F). The experimental results show that the expression of reg3 gradually increased with the prolongation of HP-CagA^+^ infection time. This indicates that CagA can promote the expression of *reg3*.

### 2.3. reg3 Regulates the Growth and Development of Tissues and Organs

The above experiments proved that *reg3* can promote cell proliferation (Figure 2C). To explore whether *reg3* could affect the growth of mice, we constructed an adeno-associated virus overexpressing *reg3* (AAV-Reg3-oe) (Figure 3A). Approximately eight weeks after AAV-Reg3-oe injection into the tail vein, the expression of *reg3* was assessed using a small animal imaging instrument (Figure 3B). The results showed that *reg3* was expressed systemically. The mice were then dissected, and the growth of the liver, lungs, and kidneys was assessed (Figure 3C–E). When *reg3* was overexpressed, the volume and weight of the liver, lung, and kidney were higher than those of the control group. These results agree with our conclusion that *reg3* promotes cell proliferation, i.e., *reg3* promotes the growth of tissues and organs by inducing cell proliferation. The level of *reg3* expression in each tissue and organ was assessed, and that of the *reg3*-oe group was determined to be upregulated (Figure 3F,G).

### 2.4. CagA and reg3 Alter the Cell Cycle

The above experiments demonstrate that CagA promotes the expression of *reg3* and stimulates cell proliferation with Reg3 (Figure 2C–F). To explore the causes of CagA and *reg3* promoting cell proliferation, cell cycle changes were assessed after the upregulation of *reg3* in gastric cancer cells and *H. pylori* infection of gastric cancer cells (Figure 4A). The results showed that the proportion of G1 phase gradually decreased with the prolongation of *reg3* overexpression. This also implies that *reg3* promotes the progression of the cell cycle. At the same time, HP-CagA^+^ was used to infect gastric cancer cells to detect cell cycle changes (Figure 4B). The results show that the proportion of G1 phase gradually decreased with the prolongation of infection. This indicates that CagA promotes the progression of the cell cycle. Combined with the above experiments, we conclude that CagA and *reg3* promote cell proliferation by altering the cell cycle. Of course, the reason for the slowdown of cell proliferation is not only the change of the cell cycle, but also the increase of the proportion of apoptosis. Therefore, the effects of *reg3* and *CagA* on apoptosis were examined (Appendix A). It was found that *reg3* and *CagA* can inhibit the apoptosis of gastric cancer cells. This article mainly analyzes changes in the cell cycle, so it does not detail the mechanism of *reg3* and *CagA* affecting apoptosis.

### 2.5. CagA and reg3 Regulate the Expression of CDK4 and CyclinD1

Both reg3 and CagA can affect the transition from G1 to S phase of the cell cycle (Figure 4). Key to the transition from G1 to S phase is the formation of cell cycle-dependent complex CDK4/cyclinD1 (Figure 5A). However, the regulatory mechanisms of CagA and *reg3* in relation to cell cycle progression are unclear. Changes in cell cycle-dependent genes *CDK4* and cyclinD1 were detected after upregulation of *reg3* in gastric cancer cells. The results show that *reg3* can promote the expression of *CDK4* and cyclinD1 (Figure 5B,D). Simultaneously, changes in the expression of *CDK4* and cyclinD1 were detected after HP-CagA+ infection of gastric cancer cells (Figure 5C,D). The results show that CagA can promote the expression of CDK4 and *CyclinD1*. The formation of the cell cycle-dependent complex CDK4/*CyclinD1* is key to the transition from cell cycle G1 to S phase. Therefore, the effect of Reg3 on the expression and distribution of CDK4 and CyclinD1 were assessed by immunofluorescence (Figure 5E). The results showed that overexpression of *reg3* increased the distribution of CDK4 and cyclinD1 in the nucleus, and co-localization was observed. Immunofluorescence was also used to detect the effects of HP-CagA+-infected cells on the expression and distribution of CDK4 and CyclinD1 (Figure 5F). The results showed that HP-CagA+ infection increased the distribution of CDK4 and CyclinD1 in the nucleus, and co-localization was observed. This may be the mechanism by which *reg3* and CagA regulate the cell cycle.

### 2.6. CagA Regulates the Expression of CDK4 and CyclinD1 through reg3

Both CagA and *reg3* can alter the cell cycle and ultimately promote cell proliferation by regulating the expression of *CDK4* and *Cyclin D1* (Figure 2, Figure 4 and Figure 5). Furthermore, CagA can promote Reg3 expression (Figure 3). So, whether CagA regulates the cell cycle and ultimately affects cell proliferation through *reg3* is a question worthy of investigation. In HP-CagA+-infected gastric cancer cells, the cell cycle was detected after downregulation and upregulation of *reg3* (Figure 6A). The results showed that when *reg3* was upregulated, CagA promoted cell cycle transition in the G1/S phase. When *reg3* was downregulated, CagA reduced the role of promoting cell cycle transition in the G1/S phase. Meanwhile, in HP-CagA+-infected gastric cancer cells, the expression of CDK4 and cyclin D1 was detected when *reg3* expression was altered (Figure 6B,C). The results showed that when *reg3* was upregulated, CagA promoted the expression of CDK4 and cyclinD1 at the protein level. When reg3 was downregulated, CagA reduced the role of promoting the expression of CDK4 and CyclinD1 at the protein level. The effects of cagA and reg3 on the mRNA level of CDK4/CyclinD1 were also examined (Figure 6D,E). We observed that changes in cagA and reg3 expression had no effect on the mRNA level of CDK4/cyclinD1. Immunofluorescence was also used to detect the effects of HP-CagA+ and reg3 on the expression and distribution of CDK4 and cyclinD1 (Figure 6F). The results showed when *reg3* was downregulated, CagA reduced the expression of CDK4 and cyclinD1 in the nucleus, and co-localization was weaker. These results suggest that *reg3* may be a key gene for CagA to alter the cell cycle and promote cell proliferation. This may also be the regulatory mechanism of *H. pylori* in promoting gastric cancer.

## 3. Discussion

The occurrence, development, and invasion of gastric cancer is a complex and continuous process. Most people with *H. pylori* infection are asymptomatic. Chronic gastritis and other diseases have occurred in only 10% of infected people [21]. However, such diseases caused by *H. pylori* infection do increase the risk of stomach cancer. Based on this evidence, the World Health Organization’s International Cancer Research Agency classified Helicobacter pylori as a Class I carcinogen in 1994 [22]. Studies have shown that cagA-positive *H. pylori* infection has an important impact on the occurrence and development of gastric cancer [11,23]. Reg3 expression influences the occurrence and development of gastric cancer [3]. In this study, immunohistochemistry was used to detect the expression of CagA and reg3 in gastric cancer and adjacent normal tissues (Figure 2A,B). We found that CagA and reg3 were upregulated in gastric cancer tissues and downregulated in the adjacent normal tissues. We concluded that CagA and reg3 play an important role in the occurrence and development of gastric cancer, and Reg3 may interact with CagA to influence the formation and development of gastric cancer. In gastric cancer cell lines, RT-PCR and Western blotting showed that the expression of reg3 significantly increased in CagA-positive *H. pylori*-infected gastric cancer cells but not in CagA-negative gastric cancer cells (Figure 2E,F). These findings suggest that reg3 may be a target molecule of CagA in gastric cancer cells.

This study has also shown that CagA and *reg3* can promote cell proliferation by affecting the cell cycle (Figure 4). We also wanted to elucidate how CagA and *reg3* regulate the cell cycle and cell proliferation. We found that CagA and *reg3* could affect the formation of the cell cycle-dependent complex, CDK4/CyclinD1 (Figure 5), which may be responsible for changes in the expression of cagA and *reg3* during the G1/S transition of the cell cycle. At the same time, we found that the regulation of the cell cycle by CagA was weaker after reg3 was downregulated (Figure 6). These findings indicate that *reg3* may be a target for CagA to regulate the cell cycle.

In summary, based on the limited number of studies on *reg3* in gastric cancer, this study elucidated the mechanism by which CagA-positive *H. pylori* can regulate the cell cycle through *reg3*, and ultimately promote the formation and development of tumors (Figure 7).

Although *H. pylori* is considered to be a class of carcinogens, it may not be a direct mutagen. CagA and its related pathogenicity island (CagPAI) are associated with increased cancer risk. CagA is transported to host cells through the type IV secretion system encoded by CagPAI, localized to the plasma membrane, and tyrosine phosphorylated by a variety of Src kinases. Tyrosine phosphorylated CagA activates SHP-2, which enhances cell proliferation. The expression of cyclin D1 induced by *H. pylori* is CagPAI dependent. It was demonstrated in this paper that CagA can regulate cyclin D1 expression through reg3, thereby affecting cell proliferation. Then, CagA’s regulation of reg3 is also cagPAI dependent. This is the question we want to study next. Generally, our research results may serve as a theoretical basis for the clinical treatment of gastric cancer.

## 4. Materials and Methods

### 4.1. Experimental Materials

Cancer tissues and corresponding adjacent tissues of gastric cancer were provided by the Military Medical University. The gastric cancer cell line MKN45 was provided by the Army Medical University. Anti-Reg3 (ab202057), anti-tubulin (ab18251), anti-CDK4 (ab108357), anti-CagA (ab224836), and anti-cyclin D1 (ab16663) were purchased from Abcam.

### 4.2. Hp-CagA Infection in Human Gastric Cancer Cells

Actively proliferating cells were plated at 2.5 × 10^5^ cells per well in a 6-well plate. When the cells are attached, replace the cell culture medium without antibiotics. The inoculating ring was used to scrape live *H. pylori* bacteria from the culture medium, suspended in phosphate-buffered saline (PBS), adjusted the bacterial density, and added the bacterial solution to the cell culture dish at a ratio of 1:100 bacteria to cells, and continued the culture. The statement of Hp strains: ATCC No. 43504; Application: Media testing Quality control strain Susceptibility testing Quality control strain for API products Enteric Research; Type Strain; Biosafety Level 2, Biosafety classification is based on U.S. Public Health Service Guidelines, it is the responsibility of the customer to ensure that their facilities comply with biosafety regulations for their own country.

### 4.3. Adeno-Related Carriers, Packaging Cells, and Strains

Virus packaging system: Three plasmids system, pAAV RC, helpers, and shuttle plasmids (carrying the target gene or shRNA). Packaging cell line: aav-293 cells, long-term preservation by hanheng biological company, culture conditions are DMEM + 10% FBS, 37 °C, 5% CO_2_, relative humidity 95%. Strain: *E. coli* strain stbl3, used to amplify adeno-associated vector and auxiliary packaging vector plasmid.

### 4.4. Gene Expression Analysis

We used UALCAN (http://ualcan.path.uab.edu) to detect reg3 expression in different tumor tissues and their corresponding normal tissues, and to compare changes in expression levels of *reg3*. UALCAN allows scientists to gain access to terabytes of data and close to 1000 reports. To systematize analyses from The Cancer Genome Atlas (TCGA) pilot and to scale their execution to the dozens of remaining diseases to be studied, UALCAN now sits atop ~40 terabytes of TCGA data and reliably executes more than 6000 pipelines per month.

### 4.5. Immunohistochemistry

Immunohistochemical steps: (1) Baking sheet: The tissue chips were placed in a 63 °C oven for 1 h. (2) Dewaxing: After baking, the tissues were placed in an automatic dyeing machine for dewaxing; (3) Antigen repair: The tissues were rinsed with pure water thrice, and then immersed in citric acid repair solution or EDTA repair solution on an electromagnetic oven and heated; (4) EDTA thermal repair: the tissues were boiled for 20 min. (5) Blocking: Commercialized ready-to-use blockers were used, and the tissues were placed onto tablets for 10 to 15 min. (6) The primary and secondary antibodies were added sequentially. (7) DAB color development: Diluted DAB was added onto the tablet and the intensity of the resulting color was assessed. (8) The tissues were stained with hematoxylin and then sealed.

### 4.6. RNA Extraction and cDNA Synthesis

Total RNA was extracted from cells using a MicroElute Total RNA kit (Omega Bio-Tek, Norcross, GA, USA). RNA from organ/tissues was extracted using TRIzol reagent (Invitrogen, USA) according to the manufacturer’s instructions. RNA quality was verified spectrophotometrically with an A260/A280 ratio between 1.8 and 2.0. cDNA was synthesized from 2 μg DNase-treated total RNA samples with oligo(dT) primers and PrimeScript™ reverse transcriptase (TaKaRa) according to the manufacturer’s instructions. cDNAs were diluted 10 times and used in qRT-RCR.

### 4.7. Fluorescence qRT-PCR

Fluorescence qRT-PCR experiments were performed on a quantitative real-time polymerase chain reaction detection system (CFX96, BIO-RAD, Hercules, CA, USA) with SYBR Premix EX Taq kit (Takara, Dalian, China), using the recommendations of the manufacturer. The primers for RT-PCR in this study were designed by Primer 5.0 and are as follows:

reg3-s: GTGACTCCTGATTGCCTCCTC, reg3-a: GCTACTCCACTCCCAACCTTC, CDK4-s: ATGGCACTTACACCCGTGGTT, CDK4-a: CTGGTCGGCTTCAGAGTTTCC, CyclinD1-s: AAGATCGTCGCCACCTGGATG, CyclinD1-a: GATGGAGTTGTCGGTGTAGATGCA, GAPDH-s: TGACTTCAACAGCGACACCCA, GAPDH-a: CACCCTGTTGCTGTAGCCAAA.

### 4.8. Cell Proliferation and Cell Cycle Test

The cell line of MKN45 was supplied by The Army Military Medical University (Chongqing, China). When the cell paved rate of more than 70%, transfected with plasmid, including empty vector (transfection reagent lip2000 come from Roche). The cells were preserved after transfection for 0, 24, 48, and 72 h, and changes in cell number and cell cycle were assessed by flow cytometry.

The MTT cytotoxicity assay was used to assess cell viability and proliferation [24]. At 48 h post-transfection, a cell suspension was prepared and loaded into 96-well culture plates. Approximately 10 µL of the MTT reagent (Beyotime Biotechnology) were added to each well and the plates were incubated for 4 h. Then, the plate cover was removed, and the absorbance of each well and the blank was measured at a wavelength of 570 nm using a microtiter plate reader.

### 4.9. Immunofluorescence

Changes in the distribution of Reg3 were determined by immunofluorescence using the following primary antibodies: Rabbit anti-Reg3 (Cell Signaling Technology). Nuclei were stained with DAPI (Beyotime Biotechnology). Fluorescent-conjugated secondary antibodies were obtained from Beyotime Biotechnology, Inc. Confocal images were collected using an Olympus TCS SP5 confocal microscope with 40×/1.25 oil objectives.

## Figures and Tables

**Figure 1 ijms-21-00224-f001:**
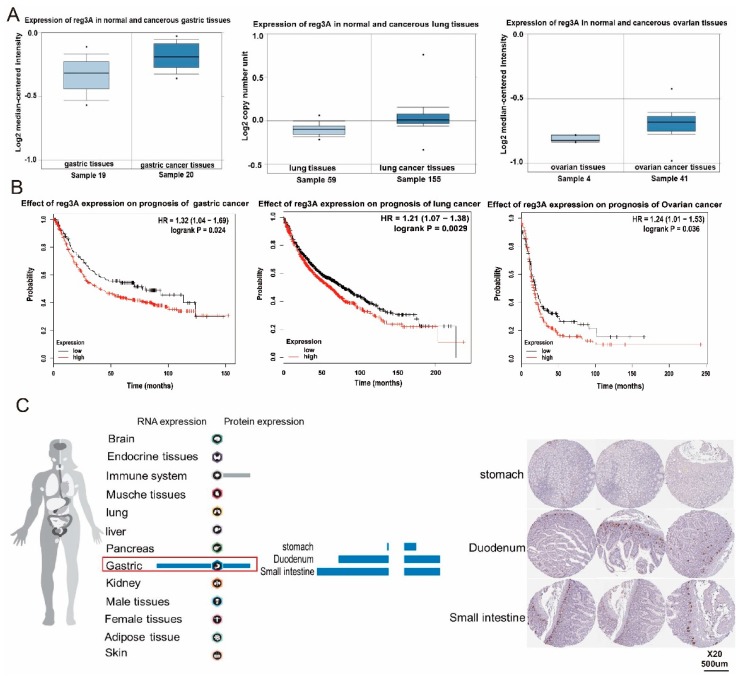
*reg3* is highly expressed in gastric cancer and affects the prognosis of patients. (**A**). The expression of *reg3* in three common tumors, including gastric, lung, and ovarian cancer. (**B**). the effect of the *reg3* on survival and prognosis of patients. (**C**). the expression of *reg3* in human tissues and organs.

**Figure 2 ijms-21-00224-f002:**
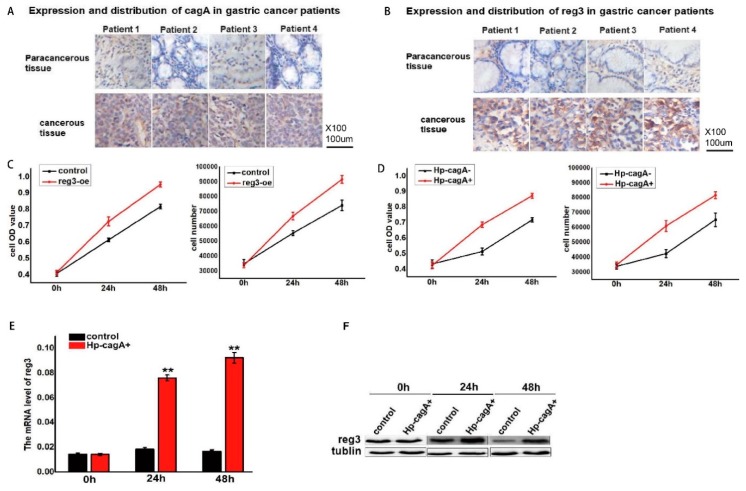
CagA promotes cell proliferation by regulating *reg3*. (**A**). Immunohistochemistry was used to detect the expression of CagA in gastric cancer. (**B**). Immunohistochemistry was used to detect the expression of reg3 in gastric cancer. (**C**). The effect of reg3 on cell proliferation was detected by the MTT assay and cell counting. (**D**). The effect of CagA on cell proliferation was detected by the MTT assay and cell counting. (**E**). The mRNA level of reg3 expression was detected at 0, 24, and 48 h after HP-CagA^+^ was transfected into gastric cancer cells. ** *p* value ≤ 0.01. (**F**). The protein level of reg3 expression was detected at 0, 24, and 48 h after HP-CagA^+^ was transfected into gastric cancer cells.

**Figure 3 ijms-21-00224-f003:**
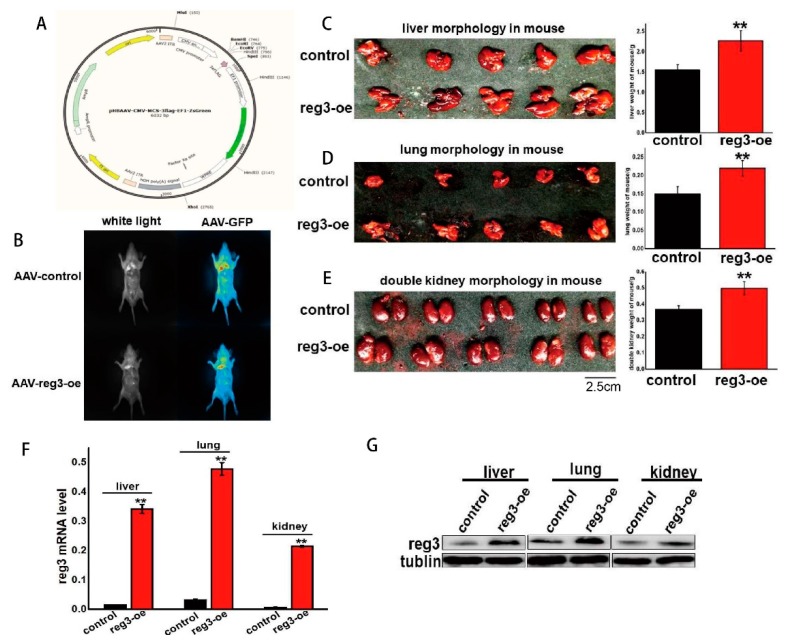
*reg3* regulates the growth and development of tissues and organs. (**A**). The viral vector map of adeno-associated virus overexpressing reg3 (AAV-Reg3-oe). (**B**). The expression of *reg3* was assessed using a small animal imaging instrument. (**C**). The mice were dissected, and the growth of the liver was assessed. (**D**). The mice were dissected, and the growth of the lungs was assessed. (**E**). The mice were dissected, and the growth of the kidneys was assessed. (**F**). The mRNA level of *reg3* in each tissue and organ was assessed. (**G**). The protein level of *reg3* in each tissue and organ was assessed. ** *p* value ≤ 0.01.

**Figure 4 ijms-21-00224-f004:**
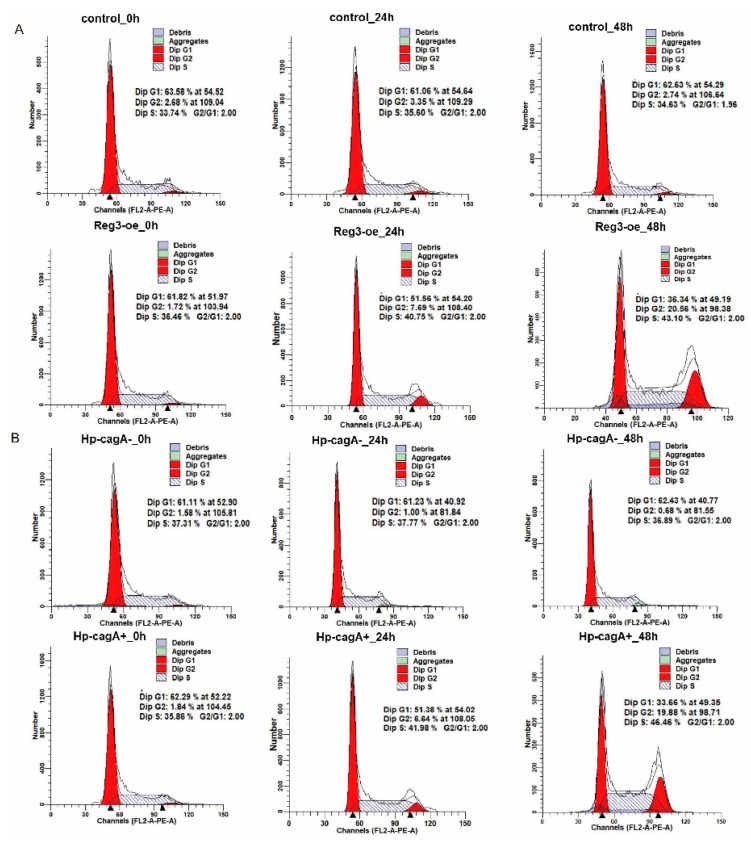
CagA and *reg3* alter the cell cycle. (**A**). cell cycle changes were assessed after the upregulation of *reg3* in gastric cancer cells. (**B**). cell cycle changes were assessed after *H. pylori* infection of gastric cancer cells.

**Figure 5 ijms-21-00224-f005:**
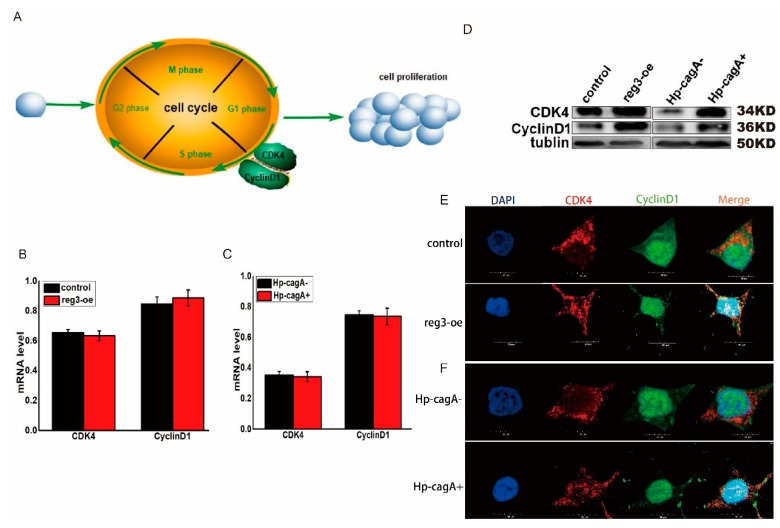
CagA and reg3 regulate the expression of CDK4 and CyclinD1. (**A**). The map of the key to G1/S phase transition of cell cycle. (**B**). The mRNA of *CDK4* and cyclinD1 were detected after upregulation of *reg3* in gastric cancer cells. (**C**). The mRNA of *CDK4* and cyclinD1 were detected after *H. pylori* infection of gastric cancer cells. (**D**). The protein level of *CDK4* and cyclinD1 were detected after upregulation of *reg3* in gastric cancer cells and *H. pylori* infection of gastric cancer cells. (**E**). The effect of Reg3 on the expression and distribution of CDK4 and CyclinD1 was assessed by immunofluorescence. (**F**). Immunofluorescence was also used to detect the effects of HP-CagA+-infected cells on the expression and distribution of CDK4 and CyclinD1.

**Figure 6 ijms-21-00224-f006:**
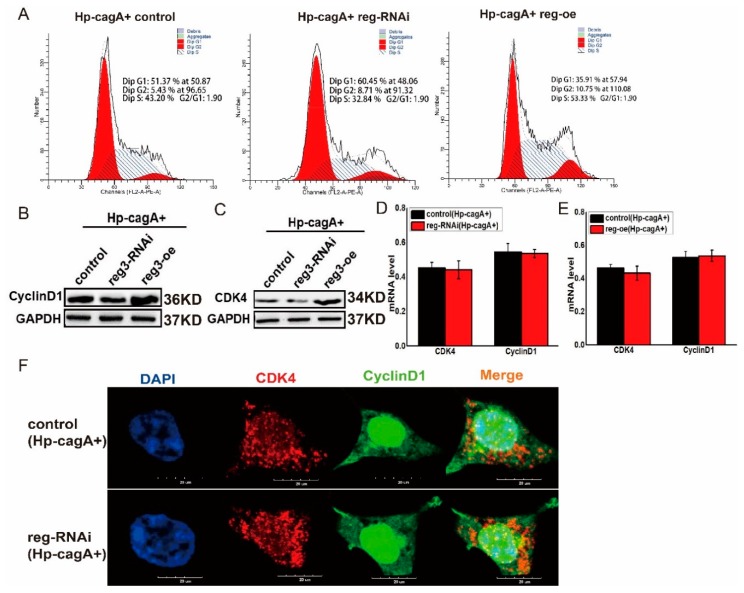
CagA and *reg3* regulate the expression of *CDK4* and *CyclinD1*. (**A**). In HP-CagA+-infected gastric cancer cells, the cell cycle was detected after downregulation and upregulation of reg3. (**B**). In HP-CagA+-infected gastric cancer cells, the expression of CDK4 was detected when reg3 expression was altered. (**C**). In HP-CagA+-infected gastric cancer cells, the expression of Cyclin D1 was detected when reg3 expression was altered. (**D**). The mRNA level of CDK4/CyclinD1 were examined after downregulation of reg3 expression in HP-CagA+-infected gastric cancer cells. (**E**). The mRNA level of CDK4/CyclinD1 were examined after upregulation of reg3 expression in HP-CagA+-infected gastric cancer cells. (**F**). Immunofluorescence was used to detect the effects of HP-CagA+ and reg3 on the expression and distribution of CDK4 and cyclinD1.

**Figure 7 ijms-21-00224-f007:**
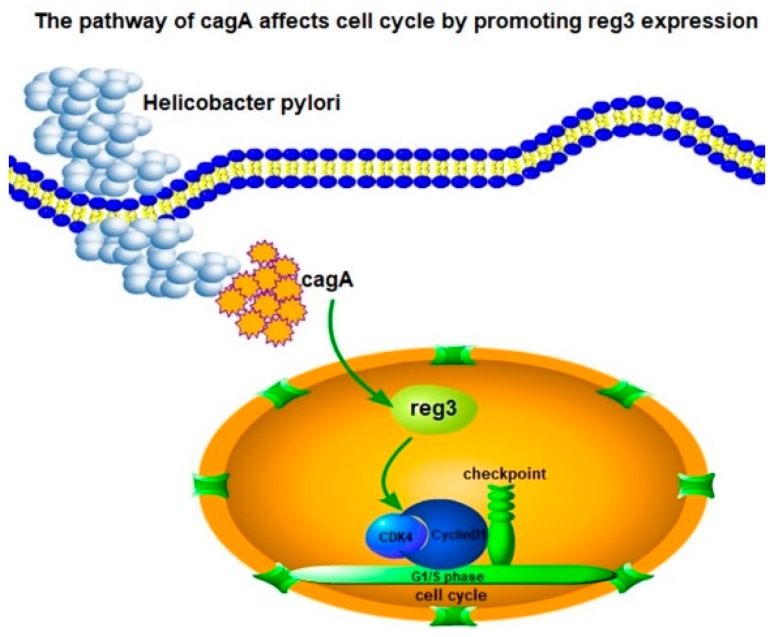
*reg3* regulates cell cycle after CagA-positive *H. pylori* infection.

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
