# Peer review of "HP-CagA+ Regulates the Expression of CDK4/CyclinD1 via reg3 to Change Cell Cycle and Promote Cell Proliferation"

_ijms, 2019, doi:10.3390/ijms21010224_

Round 1
Reviewer 1 Report
The authors examined the correlation between reg3 and H pylori-CagA and explore the effects of reg3 on the proliferation of gastric cancer cells and the development of tissues and organs. They found that reg3 may be a target gene of CagA in gastric cancer, which together affects the formation and development of gastric cancer. However, there are several concerns in this study.
Major points
Figure 1, 4, 5, and 6 are not attached in the manuscript.
Minor points
Reference should not be included in the Abstract. In Introduction, the authors stated that the five-year survival rate of gastric cancer. However, reference is too old. Reference should not be included in the Results section.4. Discussion section is too short.
Author Response
Dear Reviewer,
Thank you very much for your comments and suggestions, which were of great help in improving our manuscript. We have revised the manuscript according to your comments and suggestions.
Response to Reviewer Comments
Major points:
Point 1: Figure 1, 4, 5, and 6 are not attached in the manuscript.
Response 1: We have integrated Figure 1, 4, 5, and 6 into the manuscript.
Minor points:
Point 1: Reference should not be included in the Abstract.
Response 1: Thank you for this suggestion. Done.
Point 2: In Introduction, the authors stated that the five-year survival rate of gastric cancer. However, reference is too old.
Response 2: Thank you for this suggestion. We have updated the reference.
Point 3: Reference should not be included in the Results section.
Response 3: Thank you for this suggestion. Done.
Point 4: Discussion section is too short.
Response 4: Thank you for this suggestion. We increase the length of the manuscript to discuss the effect of H. pylori on gastric cancer formation.
Reviewer 2 Report
These are the few things the authors need to address.
The authors showed reg3 and CagA promote cell proliferation by regulating G1/S phase transition. However they do not investigate apoptosis. Changes in cell cycle/proliferation influence cell death. Investigating cell death will be an important data to add here The quality of western blots in this paper is sub-optimal, (a) the authors don't mention molecular weights, (b) western blots are often not quantified, (c) GAPDH of figure 6B has bubble in the blot, its hard to interpret data with this kind of quality, (d) the blots are cut too close to the expression of the protein detected.Author Response
Dear Reviewer,
Thank you very much for your comments and suggestions, which were of great help in improving our manuscript. We have revised the manuscript according to your comments and suggestions.
Response to Reviewer Comments
Point 1: The authors showed reg3 and CagA promote cell proliferation by regulating G1/S phase transition. However they do not investigate apoptosis. Changes in cell cycle/proliferation influence cell death. Investigating cell death will be an important data to add here.
Response 1: Thank you for this suggestion. The effects of reg3 and CagA on the apoptosis of gastric cancer cells were added in the Supplement 1 and Supplement 2.
Point 2: The quality of western blots in this paper is sub-optimal, (a) the authors don't mention molecular weights, (b) western blots are often not quantified, (c) GAPDH of figure 6B has bubble in the blot, its hard to interpret data with this kind of quality, (d) the blots are cut too close to the expression of the protein detected.
Response 1: Thank you for this suggestion. We have added the molecular weights to the pictures, and the expression of GAPDH was retested.
Round 2
Reviewer 1 Report
The authors examined the correlation between reg3 and H pylori-CagA and explore the effects of reg3 on the proliferation of gastric cancer cells and the development of tissues and organs. They found that reg3 may be a target gene of CagA in gastric cancer, which together affects the formation and development of gastric cancer. However, there are several concerns in this study.
Major points
Most important concern is that detailed method was not stated in the Method section. For example, there is no statement for anti-CagA Ab. It is not clear how Hp-CagA was transfected. There is no statement for Hp strains used. In addition, they constructed an adeno-associated virus overexpressing reg3 (AAV-Reg3-oe). However, there is no statement for detailed in the Method section. In Figure 1, the expression level of reg3 in three tumor tissues was higher than that in normal tissues. However, there is no statement in the Method section.
Minor points
Reference should not be included in the Abstract. In Introduction, the authors stated that the five-year survival rate of gastric cancer. However, reference is too old. Reference should not be included in the Results section.4. Discussion section is too short.
Author Response
Dear Reviewer,
Thank you very much for your comments and suggestions, which were of great help in improving our manuscript. We have revised the manuscript according to your comments and suggestions.
Response to Reviewer Comments
Major points:
Point 1: there is no statement for anti-CagA Ab.
Response 1: Thank you for this suggestion. We have added information about anti-CagA Ab in Materials and methods section.
Point 2: It is not clear how Hp-CagA was transfected.
Response 2: Thank you for this suggestion. We have added information about the method of Hp-CagA infection of cells in 2.2 Materials and methods section.
Point 3: There is no statement for Hp strains used.
Response 3: Thank you for this suggestion. We have added information about the statement of Hp strains in 2.2 Materials and methods section.
Point 4: In addition, they constructed an adeno-associated virus overexpressing reg3 (AAV-Reg3-oe).
Response 4: Thank you for this suggestion. We have added information about construction of AAV virus in 2.3 Materials and methods section.
Point 5: In Figure 1, the expression level of reg3 in three tumor tissues was higher than that in normal tissues. However, there is no statement in the Method section.
Response 5: Thank you for this suggestion. We have added information about the method of detecting the expression level of reg3 in 2.4 Materials and methods section.
Minor points:
Point 1: Reference should not be included in the Abstract.
Response 1: Thank you for this suggestion. Done.
Point 2: In Introduction, the authors stated that the five-year survival rate of gastric cancer. However, reference is too old.
Response 2: Thank you for this suggestion. We have updated the reference.
Point 3: Discussion section is too short.
Response 4: Thank you for this suggestion. We increase the length of the manuscript to discuss the effect of H. pylori on gastric cancer formation.
Reviewer 2 Report
Authors addressed the comments
Author Response
Dear Reviewer,
Thank you very much for your comments and suggestions, which were of great help in improving our manuscript. We have revised the manuscript according to your comments and suggestions.
Round 3
Reviewer 1 Report
This manuscript has been revised. The authors stated detailed method. However, Discussion is still too short and insufficient.
Author Response
Dear Reviewer,
Thank you very much for your comments and suggestions, which were of great help in improving our manuscript. We have revised the manuscript according to your comments and suggestions.
Response to Reviewer Comments
Point: Discussion section is too short.
Response : Thank you for this suggestion. We increase the length of the manuscript to discuss the effect of H. pylori on gastric cancer formation.